# Trans-basin Atlantic-Pacific connections further weakened by common model Pacific mean SST biases

Chen Li [1✉], Dietmar Dommenget[1] & Shayne McGregor [1]

A robust eastern Pacific surface temperature cooling trend was evident between ~1990–2013 that was considered as a pronounced contributor to the global surface warming slowdown. The majority of current climate models failed to reproduce this Pacific cooling trend, which is at least partly due to the underrepresentation of trans-basin teleconnections. Here, we investigate whether common Pacific mean sea surface temperature biases may further diminish the Atlantic-Pacific trans-basin induced Pacific cooling. Our results suggest that background Pacific SST biases act to weaken the trans-basin teleconnection by strengthening the Atlantic atmospheric stability and reducing Atlantic convection. These Pacific SST biases also act to substantially undermine the positive zonal wind-SST feedback. Furthermore, when combined, the Pacific and Atlantic SST biases led to Pacific cooling response that is almost non-existent (underestimated by 89%). Future efforts aim at reducing the model mean state biases may significantly help to improve the simulation skills of trans-basin teleconnections.

[1] ARC Centre of Excellence for Climate Extremes, School of Earth Atmosphere and Environment, Monash University, Clayton 3800 Victoria, Australia.
✉email: chen.li2@monash.edu

Tropical central-to-eastern Pacific sea surface temperature (SSTs) have experienced notable cooling during ~1990–2013, together with the strengthening of Pacific trade winds[1,2]. Pioneering studies have suggested this Pacific La-Nina-like change is a combined effect of the internal variability (e.g., Interdecadal Pacific Oscillation, IPO) and external forcing[2–7]. However, the majority of Coupled Model Intercomparison Project phase 5[8] (CMIP5) historical simulations, which are generated by perturbed initial conditions and historical anthropogenic forcing, produced a consistent Pacific warming trend in the past decades[9–11]. Further to this, the observed Pacific cooling (Fig. 1 of ref. [12]) and the Pacific trade wind intensification (Supplementary Fig. 2 of ref. [13]) are both entirely outside of the model ranges. This indicates that some model common biases may lead to underestimate the eastern Pacific cooling contribution.

The important role of the Atlantic and Indian Ocean warming trends in driving the Pacific walker circulation intensification and eastern Pacific cooling has been emphasized in many studies, while noting that the processes involved with Pacific-only SST variability do not adequately explain the unprecedented ~1990–2013 Pacific cooling event[14–19]. Previous studies[13,20,21] suggested that the Atlantic warming-induced atmospheric deep convection will trigger easterly anomaly in the Indo-western Pacific along with westerly anomaly along the eastern equatorial Pacific, reminiscent of the classic Gill-type[22] atmospheric response. The easterly wind anomaly suppresses the local convection and tends to warm the Indo-western Pacific. In the eastern Pacific, on the other hand, the Rossby-wave induced off-equatorial easterly anomaly acts to intensify the trade wind and cool the equatorial-off eastern Pacific. These SST-atmosphere interactions enhance the Pacific walker circulation with anomalous descending air in the central-to-eastern Pacific, and eventually cools the central-to-eastern Pacific through the wind-evaporation-SST effect and Bjerknes feedback[20]. A hierarchy of climate model simulations also suggests that Atlantic warming accounts for a large part of the Pacific SST cooling and related Walker circulation changes[13,20]. Thus, reproducing the trans-basin Atlantic-Pacific teleconnection is a key component for the models to capture the recent Pacific cooling trend. One previous study using targeted climate model experiments demonstrated that combining the observed Atlantic warming trend with the typical model Atlantic SST mean biases lead to a substantially underestimated Pacific cooling response[23]. This result suggested that minimizing the Atlantic mean state bias is crucial to the accurate representation of trans-basin variability[23].

In this study, we continue identifying the role of common model background SST biases in simulating the trans-basin induced Pacific cooling response to observed Atlantic warming, with a focus on the Pacific mean state bias (Supplementary Fig. 1). Our results suggest the model Pacific SST mean biases acts to further pronounced damping of the Pacific cooling response to observed Atlantic warming.

## Results

**Experiments design**. To identify the role of common model background SST biases in representing the Atlantic-Pacific teleconnection, we first conducted four sets of partially coupled (PARCP) UM7.3 (the Met Office Unified Model) experiments (Supplementary Table 1), with a slab mixed-layer ocean in the Pacific basin[24], to explore the Pacific SST response to the prescribed Atlantic warming forcing under different background SSTs, including (1) the observational global SST mean state, (2) with the CMIP5 multi-model ensemble monthly mean SST biases added into the Atlantic region, (3) with CMIP5 ensemble-mean SST biases added to the Pacific region and (4) with mean SST

biases added to both the Atlantic and Pacific regions. To achieve these differing background SSTs, flux adjustment schemes are used to mimic the observed or the biased CMIP5-like mean state in the respective Pacific Ocean basin (Supplementary Fig. 2). The fixed Atlantic warming pattern is obtained by multiplying the observed Atlantic SST trend during 1992–2011 by the total time period (Fig. 1a–d). For each set of experiments, we performed a pair of simulations: one control run where the Atlantic SST is prescribed with the climatology SST, another Atlantic warming run with additional observed Atlantic warming pattern. Thus, the difference between them can be considered as model's response to the Atlantic warming forcing (see the 'Methods' section).

**Weakened trans-basin teleconnection**. Although the Pacific cooling response occurs in all experiments, the magnitude of Pacific cooling is significantly reduced in the simulations when background SSTs in either the Pacific or Atlantic region are nudged towards the biased CMIP5 mean SST (Fig. 1b–d), relative to the observed climatology simulation (Fig. 1a). The Atlantic mean state bias appears to have somewhat stronger effect on the SST response of the eastern Pacific (Fig. 1e), while the varying SST and surface wind stress responses in the central Pacific tends to be more sensitive to the Pacific mean state bias (Fig. 1f). In Particular, the cooling magnitude over Niño3.4 region (5°S–5°N, 170°−120°W) is reduced about 32% in the Pacific bias simulation, in comparison with the unbiased simulation. A more substantial undermined cooling response (reduced about ~64%) appears in the south off-equatorial Niño3.4 region (10°S–5°S, 170°−120°W) (Supplementary Table 2). In accordance with the reduced SST cooling response, the strengthening of central Pacific trade winds (6°S–6°N, 160°E–140°W) were also largely underestimated (reduced more than 50%) in the Pacific bias simulation (Fig. 1e, f and Supplementary Table 2). Thus, simulations that utilize CMIP5-like SST background bias in the Pacific produce a robust weakening of the Pacific trade wind intensification and SST cooling in response to the prescribed Atlantic warming.

Background state biases in the Pacific and Atlantic Ocean individually both clearly act to suppress the trans-basin induced Pacific cooling. Further to this, our results also suggest that when combining the CMIP5-like background state biases in both Pacific and Atlantic Ocean basins, the simulated central-to-eastern Pacific cooling is further dampened. The combined impacts of these background state biases result in a simulated central-to-eastern Pacific cooling that is weakened by ~89% relative to the observed climatology simulation (Supplementary Table 2). This result confirms the SST mean state biases in both regions play an imperative and constructive role in reducing realistic trans-basin Atlantic-Pacific connections.

To better understand the dynamics of how the mean state biases, suppress the strengthening of Pacific trade winds and lead to the weakened Pacific cooling, we performed another four sets of atmosphere-only (AGCM) experiments. Each of these experiments is forced with prescribed climatological Pacific SSTs, that is, the SST-atmosphere coupling of the Pacific Ocean is turned off (Supplementary Table 3). This approach allows us to see the direct effect of the different background SSTs on the Pacific region's atmospheric response to the prescribed Atlantic warming in the absence of coupled interactions in the Pacific.

Comparing the AGCM experiments with the partially coupled (PARCP) experiments shows significant differences to the simulated vertical velocity in response to the imposed Atlantic Ocean warming when this is superimposed over either the observed mean state (black lines in Fig. 2), or the CMIP5-like mean states (coloured lines in Fig. 2). Results from a previous study suggest that adding the CMIP5 Atlantic SST bias, with a

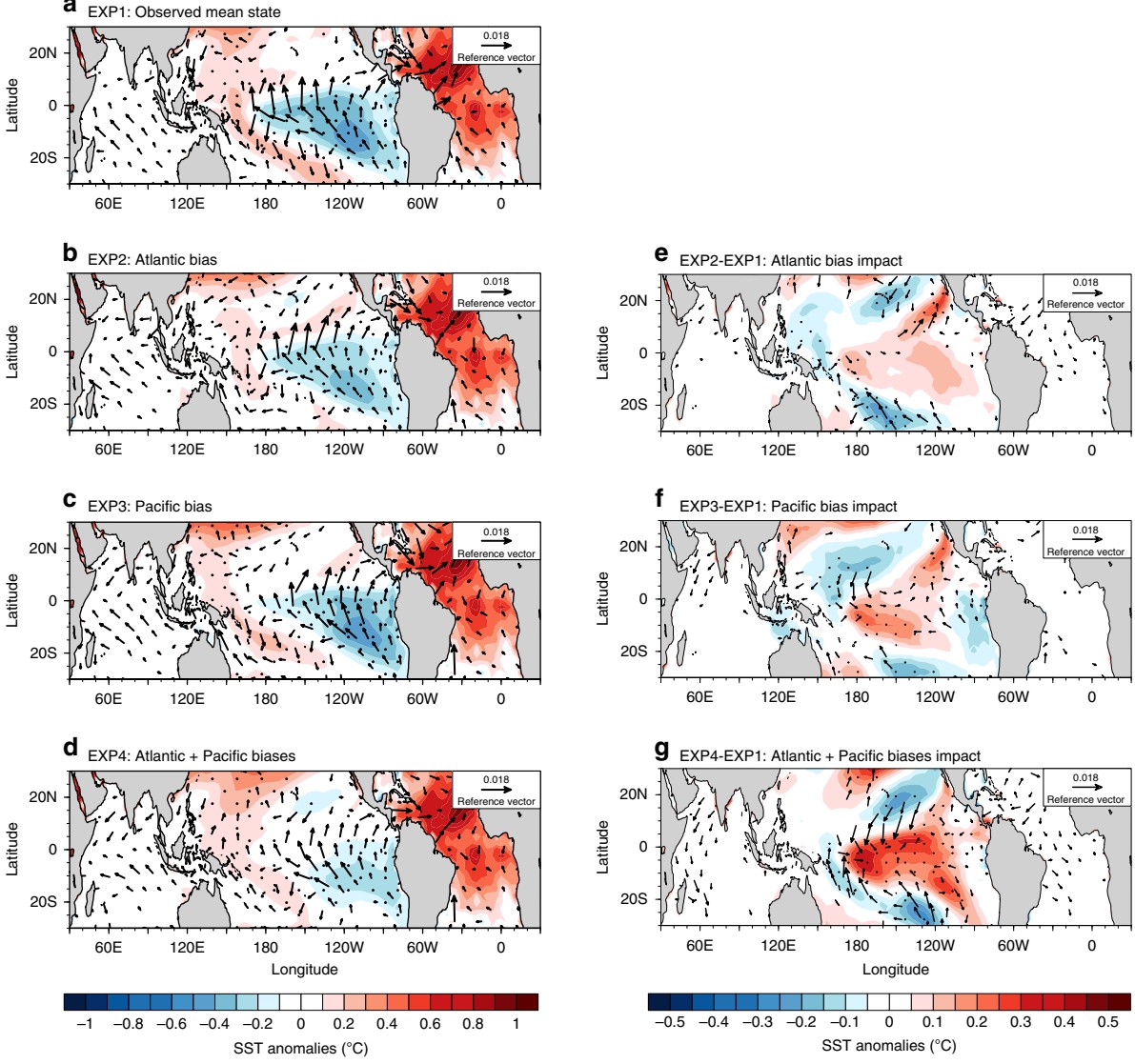

**Fig. 1 Sea surface temperature and surface wind stress response. a** Simulated sea surface temperature (SST) (shading; °C) and wind stress (vectors; N m$^{-2}$) response in the partially coupled (PARCP) simulations to the Atlantic warming forcing under the observed SST mean state (experiment 1, EXP1). **b–d** As in (**a**), but with additional CMIP5 Atlantic mean state bias (experiment 2, EXP2), Pacific bias (experiment 3, EXP3) and the combined Atlantic/Pacific bias in the SST background (experiment 4, EXP4), respectively. **e–g** The response differences between (**b**) and (**a**), (**c**) and (**a**), and (**d**) and (**a**), respectively. Stippling indicates the statistical significant at the 90% confidence level for SST. Only the significant wind stress (zonal or meridional significant at the 90% confidence level) is shown in the plots.

warm SST bias in the southeastern tropical Atlantic along with a cold bias prevailing in the northwestern Atlantic, acts to alter the regions that are above or below the threshold for deep convection[23]. This results in a weakening of the Atlantic atmospheric heating response to the prescribed tropical Atlantic warming trend along with reducing the ascending motion between 90°W and 30°W and a shift of the maximum ascending motion eastward. This weakening and eastward shift of the Atlantic heating response leads to a reduction in the descending motion of the central Pacific (180°E–135°W), further reducing the anomalous easterly wind response in the central equatorial Pacific (Fig. 2 of ref. [23]). Thus, an underestimated strengthening of Pacific Walker circulation and a weakened Pacific cooling response is expected when Atlantic warming is imposed on top of common model background SSTs in the tropical Atlantic region. Results from our Atlantic bias experiment generally confirm those of the previous study, finding a similar eastward shift of the atmospheric circulation response (Fig. 2a, c, d, f). In the remainder of this

manuscript, we will focus on exploring the impact of CMIP5 ensemble-mean background SST biases over the Pacific region.

The introduction of the Pacific region background SST bias leads to a significant reduction in upward motion over the tropical Atlantic region (Fig. 2e) in the Atmosphere-only simulations. The reduction in Atlantic ascending air along with weakened convection are also robust in the lower layers velocity potential and precipitation responses (Supplementary Fig. 3c, f). This raises the question of how biased Pacific background SSTs affect the Atlantic heating response. The CMIP5 ensemble-mean background SST shows a strong warm bias in the southeastern Pacific (Supplementary Fig. 1). This positive SST bias has been considered as a dominant driver of the so-called double intertropical convergence zone (ITCZ) problem in CMIP5 models[25], which is presented as stronger than observed precipitation in the south of the equator in the Pacific (Supplementary Fig. 4). The strong convection tends to warm the upper troposphere, in particular, the warm effect is stronger and more expanded in the upper atmosphere than the

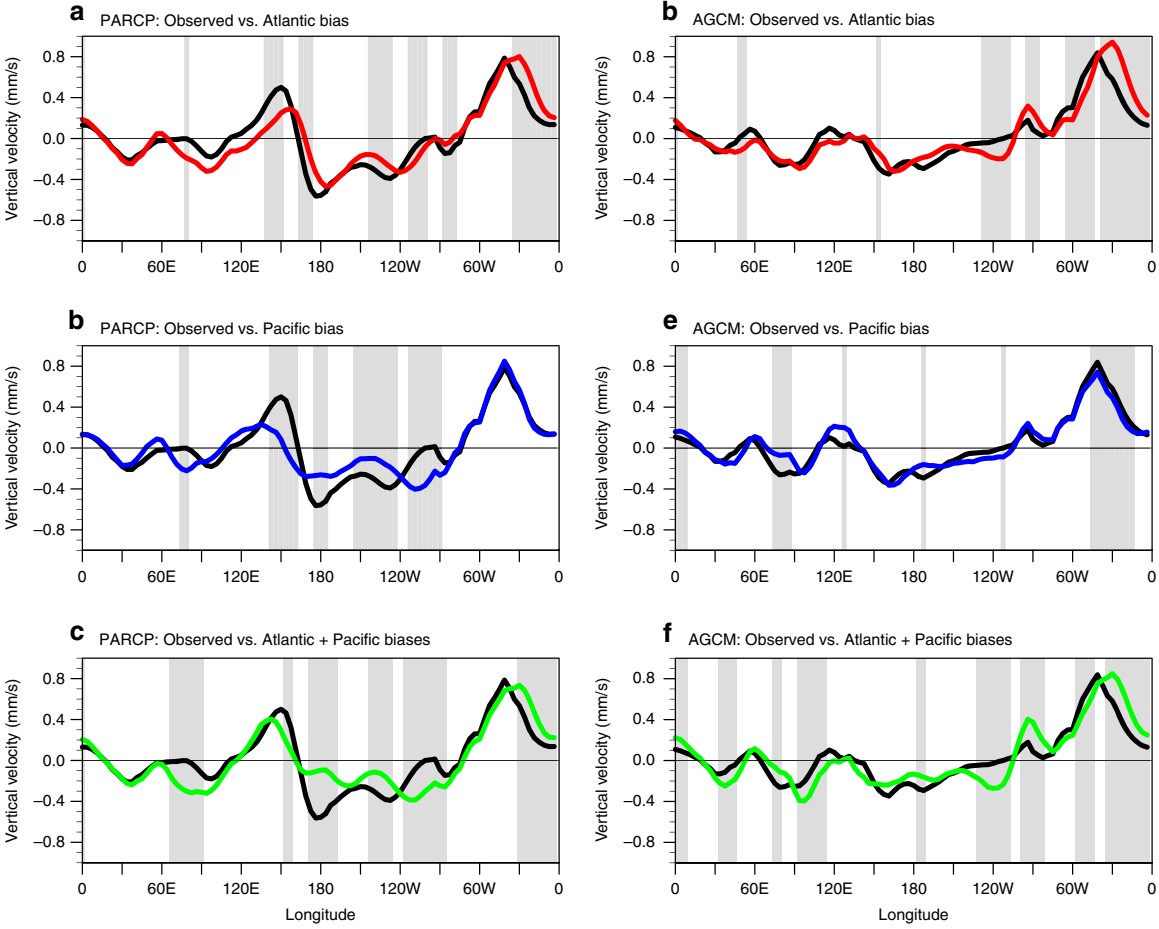

**Fig. 2 Vertical velocity response in coupled and atmosphere-only simulations. a–c** The averaged equatorial (15°S–15°N) vertical velocity (1000–200 hPa) response (unit: mm s$^{-1}$) to the Atlantic warming in the partially coupled (PARCP) simulations. The black lines represent the response under observed mean state. The red, blue and green lines represent the result from Atlantic bias, Pacific bias and combined Atlantic and Pacific bias simulations, respectively. The grey shading indicates the differences between the bias and unbiased simulations are above 90% confidence level. **d–f** As in (**a–c**), but for the atmosphere-only (AGCM) simulations. Note the positive vertical velocity represents upward motion and vice versa.

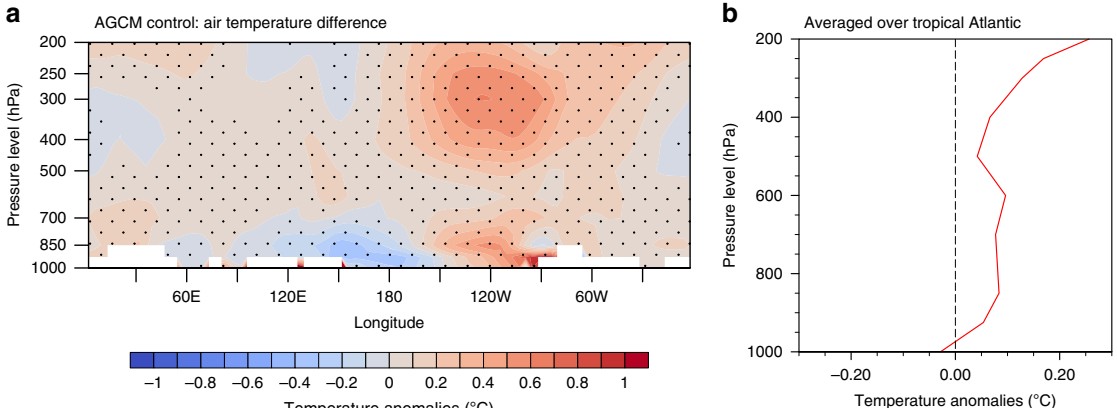

**Fig. 3 Air temperature mean state change related to the Pacific bias. a** Tropical (15°S–15°N) air temperature climatology difference between the Pacific bias and the unbiased simulations in the atmosphere-only (AGCM) control runs. Stippling indicates the statistical significance at the 90% confidence level for air temperature differences. **b** Atlantic region (60°W-0) averaged air temperature difference profile.

near-surface (Fig. 3a). This stronger warming at higher atmospheric levels stabilizes the atmosphere and reaches into tropical Atlantic, which acts to weaken the tropical Atlantic heating response to the prescribed Atlantic SST warming (Fig. 3b).

In corresponding to the weakened Atlantic heating response in the Pacific bias experiment, a reduced descending motion is shown in the central Pacific (Fig. 2b, e and Supplementary Fig. 3c, f). Note that the reduced central Pacific descending motion only becomes robust while including the surface coupling processes (c.f., Fig. 2b, e), indicates the vital role of surface heat flux changes in simulating the Atlantic-Pacific atmospheric teleconnection. The change of vertical motion can subsequently affect the surface

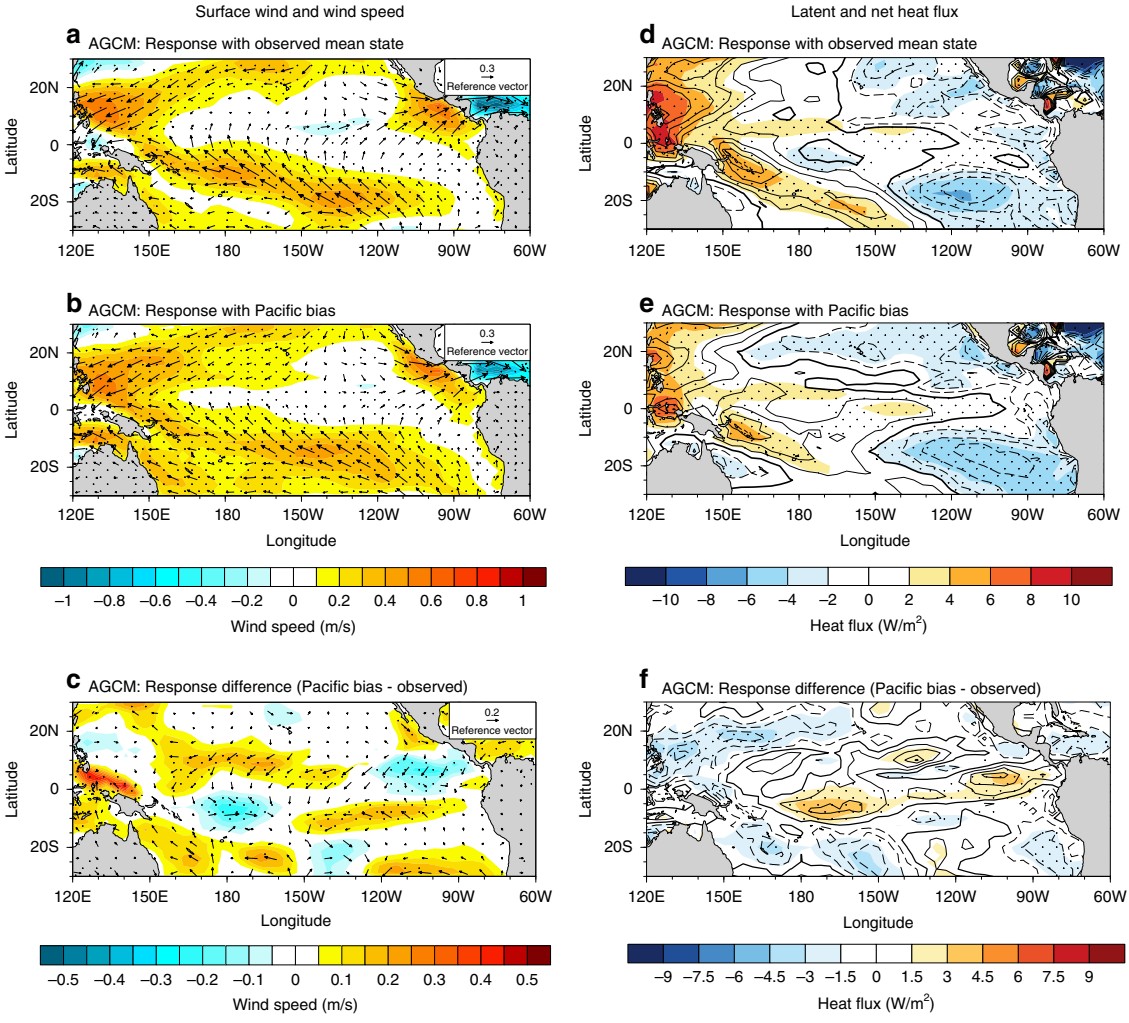

**Fig. 4 Surface wind and heat flux response in atmosphere-only simulations. a, b** Surface wind (vectors; m s$^{-1}$) and wind speed (shading; m s$^{-1}$) response to the Atlantic warming forcing in the observed mean state and Pacific bias atmosphere-only (AGCM) simulations, respectively. **c** The response differences between (**b**) and (**a**). **d–f** As in (**a–c**), but for the latent heat flux (shading; W m$^{-2}$, downward is positive) and downward net heat flux (contour; W m$^{-2}$). The solid (dashed) contour indicates positive (negative) value for net heat flux, solid bold contour indicates 0. Contour interval is 2 and 1.5 W m$^{-2}$ in (**d, e**) and (**f**), respectively. Stippling indicates the wind speed (**a–c**) and the latent heat flux (**d–f**) response or response differences are statistically significant above the 90% confidence level. Note the red (blue) colour in (**d–f**) means ocean gains (loss) latent heat flux.

heat flux through two major processes: the wind-evaporation-SST feedback and the cloud-radiation feedback[26], which appear to destructively interfere with each other. Firstly, the weaker descending velocity leads to an undermined southwesterly surface wind response (Fig. 4a–c) in the central Pacific. These surface wind changes reduce the anomalous surface wind speeds, restricting the surface latent heat flux (Fig. 4d–f), and therefore contributing to the underestimated Pacific SST cooling effect. The weakened central Pacific downward motion of this simulation is also associated with increased convection (clouds), which reduces the regional solar radiation and cools the underlying SSTs (Supplementary Fig. 5). Thus, the Pacific background SST bias acts to enhance the Pacific cooling through the cloud-radiation feedback, opposing the wind speed-latent heat flux effect. The net surface heat flux response distribution is highly consistent with the latent heat flux response (Fig. 4d–f). This suggests that the underestimated Pacific cooling trend in the Pacific bias simulation is largely contributed by the central Pacific wind-evaporation SST feedback.

**Weakened Pacific wind-SST feedback.** In addition to modulating the Atlantic-Pacific ascending and descending motions,

adding the CMIP5 Pacific background SST bias also tends to dampen the Pacific's own amplifying wind-SST feedback (the atmospheric Bjerknes feedback). This weakening feedback is manifested by a weakening relationship between the central Pacific (6°S–6°N, 160°E–140°W) zonal wind stress anomaly and the Niño3.4 SST anomaly (Fig. 5a). Consistent with previous ENSO studies[27–30], this weakened atmosphere positive feedback is associated with a weakening and westward shift Walker circulation rising branch (Fig. 5b). In observations, the strong zonal wind feedback is caused by a convective response in the western to central equatorial Pacific. A cold SST mean state in this region, which is the most common equatorial bias in the current climate models (Supplementary Fig. 1), contributes to a westward shift of Walker Circulation and a weaker convective response, and further determines the strength of zonal wind feedback[29]. It is interesting to note that the zonal wind-SST feedback in the Pacific bias simulation (EXP3) is weaker than that of in the combined Atlantic and Pacific bias simulation (EXP4), which suggests that the Atlantic bias may also influence the Pacific regions atmospheric-ocean feedback. Besides the zonal wind-SST feedback, the Pacific bias simulation with a strong cold tongue also overestimates the eastern Pacific's positive solar radiation

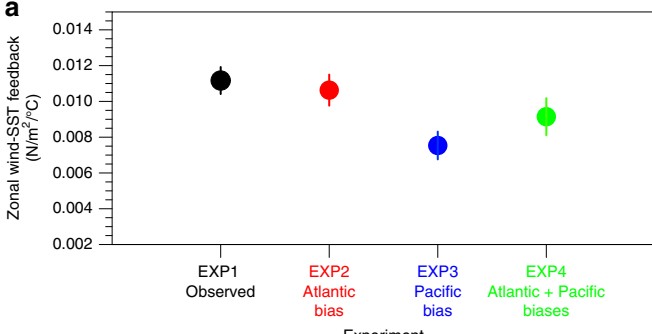

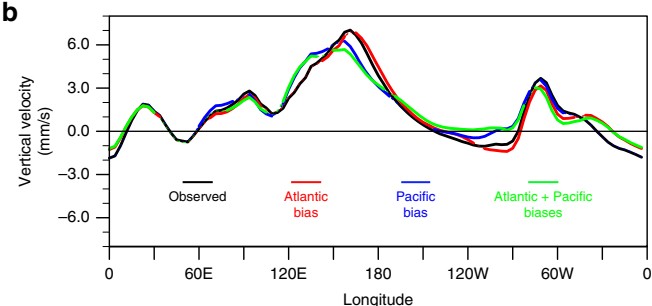

**Fig. 5 Pacific atmospheric zonal wind-SST feedback and mean Walker circulation in coupled control simulations. a** The linear regression of the monthly anomalous central Pacific zonal wind stress (6°S–6°N, 160°E–140° W) and Niño3.4 SST (5°S–5°N, 170°–120°W) under different SST mean state background (unit: N m$^{-2}$ °C$^{-1}$). Vertical bars represent the standard error of the regression coefficient. **b** Averaged equatorial (15°S–15°N) vertical velocity (200–1000 hPa) in the partially coupled (PARCP) control run simulations (unit: mm s$^{-1}$). The solid coloured lines indicate the vertical velocity differences between the biased simulations and unbiased simulation (the black line) are statistically above the 90% confidence level.

feedback[31] (Supplementary Fig. 6), which is similar to previous studies[32,33]. Our results support the idea that reducing the Pacific mean state biases will contribute to the improved simulation of Pacific atmospheric dynamics.

## Discussion

In summary, our results suggest that model Pacific background SST biases can also help to explain the underestimated Pacific cooling in response to the rapid Atlantic Ocean warming that occurred between ~1990 and 2013. The underrepresentation of this trans-basin response is largely underpinned by two prominent mechanisms: (1) the warm SST background bias in the southeastern Pacific tends to increase the atmosphere stability over the tropical Atlantic region. A subsequent Atlantic region decrease in ascending motion is found along with an associated weakening of descending air over the central Pacific. These changes lead to an underestimated Pacific trade wind strengthening, a reduced surface wind speed response, with the latter corresponding to a reduced latent heat flux (evaporation) and a weakened Pacific cooling response (Fig. 4). Thus, a weaker initial signal leading to a weaker total response, and (2) the background western to central equatorial Pacific cool SST bias acts to reduce the mean low-level convergence in the western Pacific, this led to a westward shift of Pacific walker circulation rising branch which acts to suppress the Pacific regions positive zonal wind-SST (atmospheric Bjerknes) feedback (Fig. 5). It is important to note that although the use of slab ocean model allows us to adjust the Pacific SST mean state and focus on the atmospheric dynamics, the absence of oceanic dynamics (which can provide an

amplification effect[20]) may lead to an underestimated Pacific SST response to the Atlantic forcing. Further studies are needed to address the role of SST mean state while including active ocean dynamics.

Given the importance of the future Pacific mean state to regional climate changes in precipitation[34], sea level rise[35] and ENSO changes[36], it is critical to accurately predict the future change of tropical Pacific SST under the increasing anthropogenic greenhouse gases (GHGs). Most state-of-the-art climate models predict a reduced Pacific west–east SST gradient (i.e., El Nino-like pattern) to increased GHGs forcing. In spite of this, the Pacific SST has experienced a strengthened west–east gradient (i.e., La-Nina-like pattern) in recent decades as GHGs concentrations continue sharply rising. This inconsistency is challenging our confidence in the projections of Pacific SST change. Previous studies advised that the model mean state biases can partially contribute to the uncertainty of projected tropical Pacific SST change[37,38]. Further, with the deepening understanding of the pantropical climate interactions, the model's ability in capturing the trans-basin teleconnection tends to be essential for predicting the tropical Pacific future change[39,40]. Our present study emphasises the important role of simulated SST mean state in representing the Atlantic-Pacific connection, and provides additional insights into the projected Pacific mean state change under global warming.

Results from the present study support the hypothesis that common CMIP5 surface mean state biases act to undermine the Atlantic-Pacific trans-basin variability and is partly responsible for the model underestimation of the eastern Pacific cooling trend that occurred between ~1990 and 2013. The current study is based on the assumption that the Atlantic basin warming plays the pronounced role in driving the Pacific cooling trend in spite of ongoing increases greenhouse gases forcing. However, the important role of Indian Ocean warming in modulating the Pacific climate has also been noticed[14,41]. Whether and how the mean state bias in the Indian Ocean accounts for the inconsistency between the simulated and observed Pacific decadal variability is still an open question.

## Methods

**Datasets**. The UK Met Office Hadley Centre monthly mean SST (HadISST) was employed as the observational SST. While the difference between the multi-model ensemble-mean SST from 46 CMIP5 models (the historical simulations) and HadISST was used to represent the common model mean SST biases. Monthly mean model outputs, including the SST, surface wind stress, vertical velocity, precipitation, air temperature and surface heat fluxes, are used to analysis the Pacific cooling response and related atmospheric circulation changes.

**Partially coupled (PARCP) model**. The atmosphere model, Met Office Unified Model (UM7.3), is coupled with a 50 m depth mixed-layer slab ocean in the Pacific basin. The model runs at a horizontal grid spacing of 3.75° longitude by 2.5° latitude and 38 vertical atmosphere levels.

Four sets of partially coupled experiments are performed in this study. To obtain the various Pacific (i.e., the partially coupled domain) SST background states, we estimate a flux correction that forces the model SSTs to closely follow the observed or CMIP5-like monthly mean state in the Pacific Ocean. The applied flux correction was generated using a series of iterative simulations. While in the Atlantic region (i.e., the pacemaker region), SSTs are prescribed to include the observed or CMIP5-like mean state, as detailed in the different experiments. To obtain the Pacific's response to Atlantic warming, we did one pair of simulations (i.e., the control run and the warm run) for each experiment, with the same heat flux forcing in the Pacific, but an addition prescribed Atlantic warm pattern added into the warm run. Thus, their difference can be considered as model's response to Atlantic warming under different SST backgrounds.

Note the Indian Ocean is always prescribed to the observed monthly mean SST for all the experiments.

**AGCM model**. The atmosphere model used for AGCM run is the same as the PARCP simulations (UM7.3). In comparison to the PARCP simulations, the Pacific SST is prescribed to either the observed or CMIP5-like mean state, instead of using

the flux adjustment to mimic the demanded mean state. Other settings in the Atlantic and Indian Ocean are the same as the PARCP experiments.

**Statistics information**. We used Student's *t*-test (two-sample) to estimate the significance of ensemble-mean difference at each grid between two model simulations. For each simulation, the first 10 years model outputs of the total 100 years run are excluded for the statistical analysis, thus, 90 sample sizes (represented by 90 model years) for each simulation are presumed for the significant test. We highlighted the regions that statistical significance at the 90% confidence level in the figures.

**Reporting summary**. Further information on research design is available in the Nature Research Reporting Summary linked to this article.

## Data availability

Model outputs that support the findings of this study have been deposited in https://doi.org/10.5281/zenodo.4054673.

## Code availability

The codes used to analysis the model output and generate all the figures (includes the figures in the Supplementary information) are available at https://doi.org/10.5281/zenodo.4054644.

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

## Acknowledgements

This work is supported by the ARC Centre of Excellence for Climate Extremes, Australian Research Council (Grant No. CE1700100023). S.M. was also supported by the ARC through grant number FT160100162. We acknowledge the support of staff at the National Computational Infrastructure (NCI) facility in Australia.

## Author contributions

D.D. and S.M. conceived this study, C.L. conducted the model simulations, analysed the model output and generated figures. All authors contributed to interpreting the results, writing and reviewing the manuscript.

## Competing interests

The authors declare no competing interests.

**Additional information**

