## [Peer Review File · Nature Communications]

Reviewers' Comments:

Reviewer #1:

Remarks to the Author:

Review of "Trans-basin Atlantic-Pacific connections further weakened by common model Pacific mean SST biases" by Chen Li, Dietmar Dommenges, and Shayne McGregor

This study used multiple AGCM experiments with and without thermal coupling with the ocean (i.e., a slab ocean model) to show that the warm SST biases in both the southeastern tropical Pacific and southeastern tropical Atlantic tend to damp the Atlantic-Pacific interbasin interactions. More specifically, the AGCM experiments showed that a warm tropical Atlantic produces a La Nina-like response in the Pacific. This cooling response in the Pacific is weaker when the AGCM is prescribed (or flux corrected) with the background warm SST bias in the Pacific and Atlantic basins. Further experiments and analysis suggested that the warm Pacific SST bias increases the stability of the atmosphere in the tropical Atlantic, and thus weakens the Atlantic-to-Pacific interbasin teleconnection.

This is a very nice study. The AGCM experiments showed very clearly that the warm SST bias in the tropical Pacific and Atlantic weakens the Atlantic-to-Pacific interbasin teleconnection as well summarized in Figure 1. This result has an important implication for the future projection of the tropical Pacific under the increasing anthropogenic greenhouse gases (i.e., El Nino-like or La Nina-like). So, I recommend a publication in Nature communications after a revision.

Major Comments

1) Impact of the Atlantic warm SST bias

=>

This study focused largely on the role of Pacific SST bias in the Atlantic-to-Pacific interbasin teleconnection (Figures 3 and 4, and Figures S3-S5), and provided a logical dynamic explanation. I would like to see a clear dynamic explanation as to why warm tropical Atlantic SST bias weakens the Atlantic-to-Pacific interbasin teleconnection. Is that has to do with the eastward shift in the vertical velocity? In any case, I could not find a clear dynamic explanation in this manuscript. If this was already discussed elsewhere (e.g., McGregor et al. 2018), please summarize this finding from the previous study and discuss if the new experiments agree.

2) Lines 48-49: "a slab mixed-layer ocean in the Pacific basin"

=>

This is an important limitation of this study. Please discuss if and how the absence of dynamic ocean model component may affect the results.

3) Lines 53-55: "To achieve these differing background SSTs, flux adjustment schemes are used to mimic the observed or the biased CMIP5-like mean state in the respective Pacific Ocean basin (Supplementary Fig. 2)"

=>

Different strategies are used to for the background SSTs in the Atlantic (prescribed) and Pacific (flux adjustment). Please discuss if and how the result may differ if the Atlantic background SSTs are also flux adjusted.

Li, G., Xie, S. P., Du, Y., & Luo, Y. (2016). Effects of excessive equatorial cold tongue bias on the projections of tropical Pacific climate change. Part I: The warming pattern in CMIP5 multi-model ensemble. *Climate Dynamics*, 47, 3817–3831. <https://doi.org/10.1007/s00382-016-3043-5>

4) Broader implication

The main result of this study has an important implication for the future projection of the tropical Pacific under the increasing anthropogenic greenhouse gases (i.e., El Niño-like or La Niña-like). Please discuss this border impact (or implication), toward the end of the paper, in more detail. The following references may be helpful:

Li, G., Xie, S. P., Du, Y., & Luo, Y. (2016). Effects of excessive equatorial cold tongue bias on the projections of tropical Pacific climate change. Part I: The warming pattern in CMIP5 multi-model ensemble. *Climate Dynamics*, 47, 3817–3831. <https://doi.org/10.1007/s00382-016-3043-5>

Seager, R., Cane, M., Henderson, N., Lee, D. E., Abernathy, R., & Zhang, H. (2019). Strengthening tropical Pacific zonal sea surface temperature gradient consistent with rising greenhouse gases. *Nature Climate Change*, 9(7), 517–522. <https://doi.org/10.1038/s41558-019-0505-x>

Lee, S.-K., D. Kim, G. Foltz, and H. Lopez, 2020: Pantropical response to global warming and the emergence of a La Niña-like mean state trend. *Geophys. Res. Lett.*, 47, e2019GL086497. <https://doi.org/10.1029/2019GL086497>.

Other comments

1) Lines 21-30: “Tropical central-to-eastern Pacific sea surface temperatures (SSTs) have experienced notable cooling during ~1990-2013, together with the strengthening of Pacific trade winds”
=>

The recent surface global warming hiatus has been largely attributed to IPO (PDO), which is intrinsic mode of variability. So, it is not surprising that CMIP5 historical simulation ensemble mean (or multi-model mean) cannot reproduce the hiatus (but, individual ensemble member does have IPO). As such, it is questionable if it is appropriate to use the recent surface global warming hiatus to motivate this study.

The last sentence “The observed Pacific La Niña-like change during this period has been related to both external forcing (e.g., changes in aerosols⁹ and solar radiation¹⁰), and internal variability, such as Interdecadal Pacific Oscillation (IPO)” is also out of place. It should appear earlier in this paragraph. The two references (9 and 10) discussed the potential effects of aerosol and solar activity, which are not directly related to anthropogenic global warming.

2) Lines 62-64: “Previous studies suggested that the Atlantic warming alone induces upward motion over the tropical Atlantic with descending air in the central-to-eastern Pacific Ocean, which is consistent with the Matsuno-Gill pattern”

=>

Matsuno-Gill model produces descending motion all over the global topics away from the heating source. So, it cannot explain the descending air in the central-to-eastern Pacific Ocean, in particular.

3) Lines 80-82: “Thus, a robust weakened Atlantic warming-Pacific cooling teleconnection is evident when the model simulation encloses CMIP5-like SST background bias in the Pacific.”

=>

It is hard to digest “weakened Atlantic warming-Pacific cooling teleconnection”. Perhaps, it is better to break this sentence into two.

4) Lines 86-88: “In fact, resulting in a simulated central-to-eastern Pacific cooling that is weakened by ~89% (Supplementary Table 2)”

=>

This sentence is not complete.

5) Figure 2:

=>

Did you convert the unit for the vertical velocity from Pa/sec to mm/sec? So, this is not omega?

6) Lines 108-110: "The introduction of the Pacific region background SST bias leads to a significant reduction in upward motion over the tropical Atlantic region (Fig. 2e) in the Atmosphere-only simulations"

=>

I don't see much change in the Atlantic vertical velocity between the two AGCM experiments. Perhaps, plotting the difference (EXP3 - EXP1) may help.

7) Figure S3a-c

=>

Please add divergent velocity vector.

8) Line 119: "... the warm effect is stronger and more expanded in the higher layers..."

=>

It is better to use "upper atmosphere" to replace "higher layers"

9) Line 123: "This reduced Atlantic heating response in the experiment..."

=>

This is very confusing. Please be more specific. In what experiment?

10) Lines 136-139: "The net surface heat flux response distribution is highly consistent with the latent heat flux response (Fig. 4d-f), which indicates the central Pacific wind-evaporation SST is the dominant mechanism underpinning the underestimated Pacific cooling trend in the simulation that includes CMIP5 biases background SST in the Pacific region."

=>

Please break it down into two sentences.

11) Line 146: "raising"

=>

rising

12) Line 170: "leaded to"

=>

led to

13) Lines 178-179: "However, the important role of Indian Ocean warming in modulating the Pacific climate has also been noticed"

=>

I think the following two papers should be referenced in here:

Luo, J. J., Sasaki, W., & Masumoto, Y. (2012). Indian ocean warming modulates pacific climate change. *Proceedings of the National Academy of Sciences*, 109, 18,701–18,706.
<https://doi.org/10.1073/pnas.1210239109>

Zhang, L., Han, W., Karnauskas, K. B., Meehl, G. A., Hu, A., Rosenbloom, N., & Shinoda, T. (2019). Indian Ocean warming trend reduces Pacific warming response to anthropogenic greenhouse gases: An interbasin thermostat mechanism. *Geophysical Research Letters*, 46,

10,882–10,890. <https://doi.org/10.1029/2019GL084088>.

Reviewer #2:

Remarks to the Author:

Review of the manuscript NCOMMS-20-22435-Tentitled "Trans-basin Atlantic-Pacific connections further weakened by common model Pacific mean SST biases", by Li, Dommegnet and McGregor.

This work uses AGCM and AGCM-SOM simulations to study the impact of Pacific and Atlantic biases in the simulation of the Atlantic-Pacific tropical teleconnections. The results show that the warming in the southeastern Pacific reduces the teleconnection between the Atlantic and the Pacific oceans via the increase of the atmospheric stability in the tropics.

The work is well designed and structured, and the results are relevant. I thus recommend publication of the paper after a few minor issues are taken care of:

From line 140 onwards, the authors talk about the modification Bjerknes feedback in the Pacific, but, in order to have proof of such modification, ocean dynamics have to be considered. Nevertheless, the model used in this work is a slab ocean model that, by definition only has thermodynamics included. The authors are referring to the atmospheric part of the Bjerknes feedback, but they can't really know if the ocean thermocline is going to respond to the anomalies (they can hypothesize it, but not really assert it). Moreover, adding ocean dynamics would reshape the SST-wind response in an unknown way. The authors should elaborate more on this subject in the paper.

The methodology used in PARCP simulations needs to be better explained. How do you restore to a certain climatology in the Pacific but keep the model's Pacific response to the Atlantic warm pattern?

Minor points:

Are the differences in the vertical velocities between control simulations (Figure 5b) significant?

Lines 28-30: Somehow the last sentence of the paragraph belongs in the beginning of it.

Lines 62-66: You should add Polo et al. 2015 to the references list:

Polo, I., Martin-Rey, M., Rodriguez-Fonseca, B., Kucharski, F., & Mechoso, C. R. (2015). Processes in the Pacific La Niña onset triggered by the Atlantic Niño. *Climate Dynamics*, 44(1-2), 115-131.

Line 73: Change "the cooling magnitude over Niño3.4 region (5°S-5°N, 170°-120°W) reduced" by the cooling magnitude over Niño3.4 region (5°S-5°N, 170°-120°W) is reduced".

Lines 123-125: This response is not so clear in the AGCM experiment.

Dear Editor and Reviewers,

Thank you very much for your time and constructive comments of our manuscript. We have revised the original manuscript on the basis of the reviewers' suggestions. The specific original comments of the reviewers follow, in italics, with our responses in blue. The corrected text lines and figures in the revised manuscript are shown in yellow.

Point by point response:

Reviewer #1 (Remarks to the Author):

Review of “Trans-basin Atlantic-Pacific connections further weakened by common model Pacific mean SST biases” by Chen Li, Dietmar Dommenget, and Shayne McGregor

This study used multiple AGCM experiments with and without thermal coupling with the ocean (i.e., a slab ocean model) to show that the warm SST biases in both the southeastern tropical Pacific and southeastern tropical Atlantic tend to damp the Atlantic-Pacific interbasin interactions. More specifically, the AGCM experiments showed that a warm tropical Atlantic produces a La Nina-like response in the Pacific. This cooling response in the Pacific is weaker when the AGCM is prescribed (or flux corrected) with the background warm SST bias in the Pacific and Atlantic basins. Further experiments and analysis suggested that the warm Pacific SST bias increases the stability of the atmosphere in the tropical Atlantic, and thus weakens the Atlantic-to-Pacific interbasin teleconnection.

This is a very nice study. The AGCM experiments showed very clearly that the warm SST bias in the tropical Pacific and Atlantic weakens the Atlantic-to-Pacific interbasin teleconnection as well summarized in Figure 1. This result has an important implication for the future projection of the tropical Pacific under the increasing anthropogenic greenhouse gases (i.e., El Nino-like or La Nina-like). So, I recommend a publication in Nature communications after a revision.

We thank the review very much for your helpful comments and positive recommendation. Responses to each comment are as follows.

Major Comments

1) Impact of the Atlantic warm SST bias

=>

This study focused largely on the role of Pacific SST bias in the Atlantic-to-Pacific interbasin teleconnection (Figures 3 and 4, and Figures S3-S5), and provided a logical dynamic explanation. I would like to see a clear dynamic explanation as to why warm tropical Atlantic SST bias weakens the Atlantic-to-Pacific interbasin teleconnection. Is that has to do with the eastward shift in the vertical velocity? In any case, I could not find a clear dynamic explanation in this manuscript. If this was already discussed elsewhere (e.g., McGregor et al. 2018), please summarize this finding from the previous study and discuss if the new experiments agree.

[Response 1]: Thanks for your suggestion. The impact of Atlantic mean state bias has already been discussed in McGregor et al. 2018^{Ref21}. Our Atlantic bias simulation results can largely support their findings. We have added a more detailed summary of dynamic explanation of the impact of Atlantic SST bias as:

“Results from a previous study suggest that adding the CMIP5 Atlantic SST bias, with a warm SST bias in the southeastern tropical Atlantic along with a cold bias prevailing in the northwestern Atlantic, acts to alter the regions that are above or below the threshold for deep convection²⁰. This results in a weakening of the Atlantic atmospheric heating response to the prescribed tropical Atlantic warming trend along with reducing the ascending motion between 90°W to 30°W and a shift of the maximum ascending motion eastward. This weakening and eastward shift of the Atlantic heating response leads to a reduction in the descending motion of the central Pacific (180°E to 135°W), further reducing the anomalous easterly wind response in the central equatorial Pacific (Fig. 2 of Ref²¹). Thus, an underestimated strengthening of Pacific Walker circulation and a weakened Pacific cooling response is expected when Atlantic warming is imposed on top of common model background SSTs in the tropical Atlantic region. Results from our Atlantic bias experiment generally confirm those of the previous study, finding a similar eastward shift of the atmospheric circulation response (Fig. 2a,c,d,f). In the remainder of this manuscript, we will focus on exploring the impact of CMIP5 ensemble mean background SST biases over the Pacific region.”

The above discussion can be found in the revised manuscript Lines 106-121.

2) *Lines 48-49: “a slab mixed-layer ocean in the Pacific basin”*

=>

This is an important limitation of this study. Please discuss if and how the absence of dynamic ocean model component may affect the results.

[Response 2]: Thanks for pointing this out. We understand that both the surface process and ocean dynamics are important for tropical Pacific variability. The choice of slab ocean model for the current study is based on two main reasons: (1) the simplified slab ocean allows us to adjust the Pacific mean SST to observed or CMIP5-like mean state, while it's hard to control the SST mean state in the fully coupled ocean model; (2) two previous modelling studies, which use same atmospheric GCM but coupled with slab ocean (McGregor et al. 2014^{Ref13}) and active dynamical ocean (Li et al. 2016^{Ref20}), respectively, get a similar Pacific SST response to the Atlantic warming forcing. Li et al. 2016^{Ref20} further suggested the initial SST response is first set by changes to the surface fluxes and amplified by the wind-evaporation-SST feedback. The ocean dynamical Bjerknes feedback provides a secondary amplification of the SST response. The similarity results from slab ocean and dynamical ocean model lead us to believe that the results based on the slab ocean (which can represent the surface heat flux changes and WES feedback) can provide us a good ‘first look’ to the atmosphere teleconnection, albeit the Pacific SST response magnitude may change if ocean dynamics are included.

We have noted the limitation of slab ocean in the revised manuscript as:

“It is important to note that although the use of slab ocean model allows us to adjust the Pacific SST mean state and focus on the atmospheric dynamics, the absence of oceanic dynamics (which can provide an amplification effect²⁰) may lead to an underestimated Pacific SST response to the Atlantic forcing. Further studies are needed to address the role of SST mean state while including active ocean dynamics.” Please refer to Lines 187-191.

3) Lines 53-55: “To achieve these differing background SSTs, flux adjustment schemes are used to mimic the observed or the biased CMIP5-like mean state in the respective Pacific Ocean basin (Supplementary Fig. 2)”

=>

Different strategies are used to for the background SSTs in the Atlantic (prescribed) and Pacific (flux adjustment). Please discuss if and how the result may differ if the Atlantic background SSTs are also flux adjusted.

[Response 3]: We have followed two different approaches for the Atlantic and the Pacific, as we are interested in different aspects for the Atlantic and the Pacific. For this study we treat the Atlantic as a pacemaker, and we are interested in the Pacific response to the Atlantic forcing.

While it may be possible to conduct such a pacemaker experiment with heat flux corrections (i.e., the flux correction approach is able to largely represent the SST background (as shown in Supplementary Fig. 2 a, b) and simulate the warm trend pattern in the Atlantic), it would be more complicated compared with the traditional pacemaker experiments (with fixed SST). To our knowledge this has not been done in most (or any) previous pacemaker experiments. The different approaches may lead to slightly differences of the SST background in the pacemaker region (Atlantic). However, the present study focuses on the Pacific response, thus, we believe that using flux adjustment instead of prescribing the Atlantic background SST would have limited impact to our main results.

4) Broader implication

The main result of this study has an important implication for the future projection of the tropical Pacific under the increasing anthropogenic greenhouse gases (i.e., El Nino-like or La Nina-like). Please discuss this border impact (or implication), toward the end of the paper, in more detail. The following references may be helpful:

Li, G., Xie, S. P., Du, Y., & Luo, Y. (2016). Effects of excessive equatorial cold tongue bias on the projections of tropical Pacific climate change. Part I: The warming pattern in CMIP5 multi-model ensemble. *Climate Dynamics*, 47, 3817–3831. <https://doi.org/10.1007/s00382-016-3043-5>

Seager, R., Cane, M., Henderson, N., Lee, D. E., Abernathey, R., & Zhang, H. (2019). Strengthening tropical Pacific zonal sea surface temperature gradient consistent with rising greenhouse gases. *Nature Climate Change*, 9(7), 517–522. <https://doi.org/10.1038/s41558-019-0505-x>

Lee, S.-K., D. Kim, G. Foltz, and H. Lopez, 2020: Pantropical response to global warming and the emergence of a La Nina-like mean state trend. *Geophys. Res. Lett.*, 47, e2019GL086497. <https://doi.org/10.1029/2019GL086497>.

[Response 4]: Thank you for providing these references, they are very helpful. We added a paragraph to discuss the implication of our results to the future projection of tropical Pacific as:

“Given the importance of the future Pacific mean state to regional climate changes in precipitation (Xie et al. 2010^{Ref34}), sea level rise (Timmermann et al. 2010^{Ref35}) and ENSO changes (Power et al. 2013^{Ref36}), it is critical to accurately predict the future change of tropical Pacific SST under the increasing anthropogenic greenhouse gases (GHGs). Most state-of-the-art climate models predict a reduced Pacific west-east SST gradient (i.e., El

Nino-like pattern) to increased GHGs forcing. In spite of this, the Pacific SST has experienced a strengthened west-east gradient (i.e., La Nina-like pattern) in recent decades as GHGs concentrations continue sharply rising. This inconsistency is challenging our confidence in the projections of Pacific SST change. Previous studies advised that the model mean state biases can partially contribute to the uncertainty of projected tropical Pacific SST change (Li et al. 2016^{Ref37}; Seager et al. 2019^{Ref38}). Further, with the deepening understanding of the pantropical climate interactions, the model's ability in capturing the trans-basin teleconnection tends to be essential for predicting the tropical Pacific future change (Cai et al. 2019^{Ref39}; Lee et al. 2020^{Ref40}). Our present study emphasises the important role of simulated SST mean state in representing the Atlantic-Pacific connection, and provides additional insights into the projected Pacific mean state change under global warming." Please refer to **Lines 192-205** in the revised manuscript.

Other comments

1) Lines 21-30: *"Tropical central-to-eastern Pacific sea surface temperatures (SSTs) have experienced notable cooling during ~1990-2013, together with the strengthening of Pacific trade winds"*

=>

The recent surface global warming hiatus has been largely attributed to IPO (PDO), which is intrinsic mode of variability. So, it is not surprising that CMIP5 historical simulation ensemble mean (or multi-model mean) cannot reproduce the hiatus (but, individual ensemble member does have IPO). As such, it is questionable if it is appropriate to use the recent surface global warming hiatus to motivate this study.

[Response 1a]: Thank you for your helpful advice. Yes, we agree with the reviewer that we cannot expect the CMIP5 multi-model ensemble mean to reproduce the Pacific cooling trend ~1990-2013, as it is largely related to the natural variability (i.e., IPO/PDO). But we may expect this observed cooling trend to be presented inside of the model range. However, none of the CMIP5 historical simulations can reproduce the recent eastern Pacific cooling trend (Fig. 1 of Luo et al. 2017^{Ref12}) as well as the Pacific trade wind intensification (Supplementary Fig. 2 of McGregor et al. 2014^{Ref13}), suggesting the CMIP5 models may underestimate the eastern Pacific cooling contribution due to some common model biases.

The last sentence "The observed Pacific La Niña-like change during this period has been related to both external forcing (e.g., changes in aerosols⁹ and solar radiation¹⁰), and internal variability, such as Interdecadal Pacific Oscillation (IPO)" is also out of place. It should appear earlier in this paragraph. The two references (9 and 10) discussed the potential effects of aerosol and solar activity, which are not directly related to anthropogenic global warming.

[Response 1b]: We have moved this sentence ahead and modified the references.

We have modified our first paragraph as:

"Tropical central-to-eastern Pacific sea surface temperature (SSTs) have experienced notable cooling during ~1990-2013, together with the strengthening of Pacific trade winds^{1,2}. Pioneering studies have suggested this Pacific La-Nina like change is a combined effect of the internal variability (e.g., Interdecadal Pacific Oscillation, IPO) and external forcing (England et al. 2014^{Ref2}; Meehl et al. 2013^{Ref3}; Kosaka and Xie 2013^{Ref4}; Marotzke and Forster 2015^{Ref5}; Takahashi and Watanabe 2016^{Ref6}; Smith 2016^{Ref7}). However, the majority

of Coupled Model Intercomparison Project phase 5⁸ (CMIP5) historical simulations, which are generated by perturbed initial conditions and historical anthropogenic forcing, produced a consistent Pacific warming trend in the past decades⁹⁻¹¹. Further to this, the observed Pacific cooling (Fig. 1 of Ref¹²) and Pacific trade wind intensification (Supplementary Fig. 2 of Ref¹³) are both entirely outside of the model ranges. This indicates that some model common biases may lead to underestimate the eastern Pacific cooling contribution.” Please refer to Lines 21-31.

2) *Lines 62-64: “Previous studies suggested that the Atlantic warming alone induces upward motion over the tropical Atlantic with descending air in the central-to-eastern Pacific Ocean, which is consistent with the Matsuno-Gill pattern”*

=>

Matsuno-Gill model produces descending motion all over the global tropics away from the heating source. So, it cannot explain the descending air in the central-to-eastern Pacific Ocean, in particular.

[Response 2]: Thanks for the suggestion. Yes, the Matsuno-Gill model produces descending motion all over the global tropical away from the heating source, but not with the same rate of descent everywhere. We agreed with the reviewer that it’s improper to directly connect the descending air in central-to-eastern Pacific with the Atlantic warming Gill-pattern. In the revised manuscript, we explained how the Atlantic warming causes eastern Pacific cooling as following:

“Previous studies (McGregor et al. 2014^{Ref13}; Li et al. 2016^{Ref20}; Polo et al. 2014^{Ref23}) suggested that the Atlantic warming induced atmospheric deep convection will trigger easterly anomaly in the Indo-western Pacific along with westerly anomaly along the eastern equatorial Pacific, reminiscent of the classic Gill-type²⁴ atmospheric response. The easterly wind anomaly suppresses the local convection and tends to warm the Indo-western Pacific. In the eastern Pacific, on the other hand, the Rossby-wave induced off-equatorial easterly anomaly acts to intensify the trade wind and cool the equatorial-off eastern Pacific. These SST-atmosphere interactions enhance the Pacific walker circulation with anomalous descending air in the central-to-eastern Pacific, and eventually cools the central-to-eastern Pacific through the wind-evaporation-SST effect and Bjerknes feedback²⁰.” Please refer to Lines 63-71.

3) *Lines 80-82: “Thus, a robust weakened Atlantic warming-Pacific cooling teleconnection is evident when the model simulation encloses CMIP5-like SST background bias in the Pacific.”*

=>

It is hard to digest “weakened Atlantic warming-Pacific cooling teleconnection”. Perhaps, it is better to break this sentence into two.

[Response 3]: We revised this sentence as:

“Thus, simulations that utilize CMIP5-like SST background bias in the Pacific produce a robust weakening of the Pacific trade wind intensification and SST cooling in response to the prescribed Atlantic warming.” Please refer to Lines 85-87.

4) *Lins 86-88: “In fact, resulting in a simulated central-to-eastern Pacific cooling that is weakened by ~89% (Supplementary Table 2)”*

=>

This sentence is not complete.

[Response 4]: Thanks for pointing this out. We have completed this sentence as: “The combined impacts of these background state biases result in a simulated central-to-eastern Pacific cooling that is weakened by ~89% relative to the observed climatology simulation.” Please refer to Lines 91-94.

5) Figure 2:

=>

Did you convert the unit for the vertical velocity from Pa/sec to mm/sec? So, this is not omega?

[Response 5]: Yes, it is not omega. The model output we used here is called “upward air velocity” on pressure levels with unit mm/sec. Thus, the positive value represents upward motion and vice versa. We have added a detailed description in the Fig.2 caption. Please refer to Lines 394-395.

6) Lines 108-110: *“The introduction of the Pacific region background SST bias leads to a significant reduction in upward motion over the tropical Atlantic region (Fig. 2e) in the Atmosphere-only simulations”*

=>

I don't see much change in the Atlantic vertical velocity between the two AGCM experiments. Perhaps, plotting the difference (EXP3 - EXP1) may help.

[Response 6]: Thank you for your suggestion. In the Fig.S3 a-c, we show the low-layers (950-500hPa) averaged velocity potential response, which indicates the vertical velocity change. We can see a significant reduced upward motion over the Atlantic region in the difference plot Fig.S3 (c) (EXP3-EXP1; Pacific bias – observed mean state). Please refer to the supplementary Figure 3c.

7) Figure S3a-c

=>

Please add divergent velocity vector.

[Response 6]: We have added the divergent velocity vector in FigS3a-c.

8) Line 119: *“... the warm effect is stronger and more expanded in the higher layers...”*

=>

It is better to use “upper atmosphere” to replace “higher layers”.

[Response 8]: We have changed to “upper atmosphere”. Please refer to Line 132.

9) *Line 123: “This reduced Atlantic heating response in the experiment....”*

=>

This is very confusing. Please be more specific. In what experiment?

[Response 9]: We have changed this sentence into:

“In corresponding to the weakened Atlantic heating response in the Pacific bias experiment, a reduced descending motion is shown in the central Pacific (Fig. 2b, e and Supplementary Fig.3c, f). Please refer to Lines 136-138.”

10) *Lines 136-139: “The net surface heat flux response distribution is highly consistent with the latent heat flux response (Fig. 4d-f), which indicates the central Pacific wind-evaporation SST is the dominant mechanism underpinning the underestimated Pacific cooling trend in the simulation that includes CMIP5 biases background SST in the Pacific region.”*

=>

Pls break it down into two sentences.

[Response 10]: We have changed this sentence into:

“The net surface heat flux response distribution is highly consistent with the latent heat flux response (Fig. 4d-f). This suggests that the underestimated Pacific cooling trend in the Pacific bias simulation is largely contributed by the central Pacific wind-evaporation SST feedback.” Please refer to Lines 151-154.

11) *Line 146: “raising”*

=>

rising

12) *Line 170: “leaded to”*

=>

led to

[Response 11,12]: Thank you for your careful reading. We have corrected them. Please refer to Line 161 and Line 185.

13) *Lines 178-179: “However, the important role of Indian Ocean warming in modulating the Pacific climate has also been noticed”*

=>

I think the following two papers should be referenced in here:

Luo, J. J., Sasaki, W., & Masumoto, Y. (2012). Indian ocean warming modulates pacific climate change. *Proceedings of the National Academy of Sciences*, 109, 18,701–18,706. <https://doi.org/10.1073/pnas.1210239109>

Zhang, L., Han, W., Karnauskas, K. B., Meehl, G. A., Hu, A., Rosenbloom, N., & Shinoda, T. (2019). Indian Ocean warming trend reduces Pacific warming response to anthropogenic greenhouse gases: An interbasin thermostat mechanism. *Geophysical Research Letters*, 46, 10,882–10,890. <https://doi.org/10.1029/2019GL084088>.

[Response 13]: Thanks for your suggestions, the two references have been included in our revised manuscript. Please refer to Line 212.

Reviewer #2 (Remarks to the Author):

Review of the manuscript NCOMMS-20-22435-Tentitled “Trans-basin Atlantic-Pacific connections further weakened by common model Pacific mean SST biases”, by Li, Dommegnet and McGregor.

This work uses AGCM and AGCM-SOM simulations to study the impact of Pacific and Atlantic biases in the simulation of the Atlantic-Pacific tropical teleconnections. The results show that the warming in the southeastern Pacific reduces de teleconnection between the Atlantic and the Pacific oceans via the increase of the atmospheric stability in the tropics.

The work is well designed and structured, and the results are relevant. I thus recommend publication of the paper after a few minor issues are taken care off:

Thank you for the very positive recommend. In the revised manuscript, we have included a discussion of the potential impacts of ocean dynamics and added more explanations for the PARCP experiments. For details, please see the following point-to-point response.

1) From line 140 onwards, the authors talk about the modification Bjercknes feedback in the Pacific, but, in order to have proof of such modification, ocean dynamics have to be considered. Nevertheless, the model used in this work is a slab ocean model that, by definition only has thermodynamics included. The authors are referring to the atmospheric part of the Bjercknes feedback, but they can't really know if the ocean thermocline is going to respond to the anomalies (they can hypothesize it, but not really assert it). Moreover, adding ocean dynamics would reshape the SST-wind response in an unknown way. The authors should elaborate more on this subject in the paper.

[Response 1]: Thank you for the suggestion. We have double checked the use of word atmospheric Bjercknes feedback, and make sure it clearly refers to the zonal wind - SST feedback only.

Besides, we have added a discussion of the potential impact of the ocean dynamics as: “It is important to note that although the use of slab ocean model allows us to adjust the Pacific SST mean state and focus on the atmospheric dynamics, the absent of oceanic dynamics (which can provide an amplification effect²⁰) may lead to an underestimated Pacific SST response to the Atlantic forcing. Further studies are needed to address the role of SST mean state while including active ocean dynamics.”

Please refers to Lines 187-191.

2) The methodology used in PARCP simulations needs to be better explained. How do you restore to a certain climatology in the Pacific but keep the model's Pacific response to the Atlantic warm pattern?

[Response 2]: Thanks for your suggestions. We have added more detailed information of PARCP run in the Method part, as following:

“Four sets of partially coupled experiments are performed in this study. To obtain the various Pacific (i.e., the partially coupled domain) SST background states, we estimate a flux correction that forces the model SSTs to closely follow the observed or CMIP5-like monthly mean state in the Pacific Ocean. The applied flux correction was generated using a series of iterative simulations. While in the Atlantic region (i.e., the pacemaker region), SSTs are prescribed to include the observed or CMIP5-like mean state, as detailed in the different

experiments. To obtain the Pacific’s response to Atlantic warming, we did one pair of simulations (i.e., the control run and the warm run) for each experiment, with the same heat flux forcing in the Pacific, but an additional prescribed Atlantic warm pattern added into the warm run. Thus, their difference can be considered as model’s response to Atlantic warming under different SST backgrounds.”

Please refer to Lines 228-238.

Minor points:

1) Are the differences in the vertical velocities between control simulations (Figure 5b) significant?

[Response 1]: Yes, the weakened Walker circulation rising branch is significant when including the Pacific mean state bias. We have added the significant test for Fig. 5b, as following:

Fig. 5(b) Averaged equatorial (15S-15N) vertical velocity (200-1000hPa) in the PARCP control run simulation. The solid colored lines indicate the vertical velocity differences between the biased simulations and unbiased simulation (the black line) are statistically above the 90% level.

Please refer to Fig.5 Lines 427-429.

2) Lines 28-30: Somehow the last sentence of the paragraph belongs in the beginning of it.

[Response 2]: We have moved this sentence ahead, please refer to Lines 23-24.

3) Lines 62-66: You should add Polo et al. 2015 to the references list:

Polo, I., Martin-Rey, M., Rodriguez-Fonseca, B., Kucharski, F., & Mechoso, C. R. (2015). Processes in the Pacific La Niña onset triggered by the Atlantic Niño. *Climate Dynamics*, 44(1-2), 115-131.

[Response 3]: Thanks for your suggestion, we have added this reference (Ref²³) when discussing the mechanism of ‘Atlantic induced Pacific change’. Please refer to Line 63.

4) Line 73: Change “the cooling magnitude over Niño3.4 region (5°S-5°N, 170°-120°W) reduced” by the cooling magnitude over Niño3.4 region (5°S-5°N, 170°-120°W) is reduced”.

[Response 4]: Thank you for your careful reading. We have corrected this sentence, please refer to Line 79.

Lines 123-125: This response is not so clear in the AGCM experiment.

[Response 5]: Thanks for pointing this out. Yes, the reduced central Pacific descending motion in the Pacific bias AGCM simulation is much weaker but still significant for a small region near the dateline, in comparison with the coupled simulation. We think this is reasonable since the wind-evaporation-SST feedback, which is a dominant mechanism for the enhancement of Pacific Walker circulation, is absent in the AGCM simulation.

In the revised manuscript, we have pointed out this difference between the PARCP and AGCM simulations as:

“Note that the reduced central Pacific descending motion only becomes robust while including the surface coupling processes (c.f., Fig. 2b and Fig. 2e), indicates the vital role of surface heat flux changes in simulating the Atlantic-Pacific atmospheric teleconnection.”

Please refer to Lines 138-140.

Reviewers' Comments:

Reviewer #1:

Remarks to the Author:

The revised manuscript adequately addressed my major comments. So, I recommend this paper be published in Nature Communications.

Reviewer #2:

Remarks to the Author:

The authors have addressed my concerns, although I still don't like the term "atmospheric Bjerknnes feedback", it is true that the term has been already used in the literature. I think that the paper is now suitable for publication after a very few minor points:

Lines 64 to 73: This paragraph must be placed before in the manuscript, I think it fits better right after line 42.

Line 64: Although I suggested to include Polo et al., 2015 reference in the previous manuscript, I really don't like how it is cited in line 64. Polo et al. deals with the Atlantic impact on the Pacific at interannual timescales, while the other two references (McGregor et al. 2014 and Li et al., 2016) deal with trends. Either differentiate the two points of view or remove again Polo et al. reference.

Line 86: Change "were" to "was".

Line 191: Change "(atmospheric Bjerknnes) feedback" to "(atmospheric Bjerknnes feedback)"

Reviewer #1 (Remarks to the Author):

The revised manuscript adequately addressed my major comments. So, I recommend this paper be published in Nature Communications.

We highly appreciate the reviewers' time and efforts in reviewing our manuscript. Thank you very much for your constructive suggestions and comments, which helped us to improve the quality of the manuscript.